# Effects of Caffeine and Caffeinated Beverages in Children, Adolescents and Young Adults: Short Review

**DOI:** 10.3390/ijerph182312389

**Published:** 2021-11-25

**Authors:** Rita Soós, Ádám Gyebrovszki, Ákos Tóth, Sára Jeges, Márta Wilhelm

**Affiliations:** 1Doctoral School of Health Sciences, Faculty of Health Sciences, University of Pécs, H-7621 Pécs, Hungary; jegessara@gmail.com; 2Institute of Sport Sciences and Physical Education, Faculty of Science, University of Pécs, H-7624 Pécs, Hungary; gyebrovszkiadam@gmail.com (Á.G.); tothahu@gmail.com (Á.T.); mwilhelm@gamma.ttk.pte.hu (M.W.)

**Keywords:** caffeine, energy drink, alcohol, children, young adults, side effects

## Abstract

The prevalence of ED consumption has increased over the past 10–15 years. Studies describing the effects of caffeine and caffeinated beverages show confusing results, so it seems important to regularly summarize the available facts, and in more detail. By a thorough analysis of more than 156 scientific papers, the authors describe the molecular background of absorption, as well as the positive and negative effects of different dosages of caffeine, just like its effects in physical activity and performance. ED and EDwA consumption is a regular habit of not only adults, but nowadays even of children and adolescents. There are no safe dosages described of caffeine or ED consumption for children. There are no positive short- or long-term effects of these compounds/products concerning developing brain functions, psycho-motor functions, or social development. Instead, there are many unpleasant side effects, and symptoms of regular or higher-dose ED consumption, especially at younger ages. This mini review describes many details of these unpleasant side effects, their severity, and motivations for consuming these compounds/products. In a quantitative research in Hungary (10–26 years, mean age: 15.6 ± 3.8 y, 1459 subjects, randomly chosen population), a survey based on a questionnaire asking people about their ED consumption habits was conducted. According to the data, 81.8% of the participants consumed EDs at least once, and 63.3% tried several products of the kind. A positive correlation was found between age and consumption (*p* < 0.001). The results show that a high proportion of this group often consumed EDwA, in many cases leading to harmful side-effects of caffeine overdose. In a sample of Hungarian high school and college students (17–26 years), ED consumption matched the international data, and only 19.7% of respondents did not use EDs at all (had never tasted an ED in their life).

## 1. Introduction

Caffeine is a compound of many plants—a special psychostimulant, according to some data—and it is one of the most frequently used psychoactive substances in the world [1]. After consumption, it is absorbed quickly.

The potential cellular effects of caffeine might be explained by three mechanisms:

The antagonism of adenosine receptors (especially in the central nervous system/CNS) [2,3]; mobilization of the intracellular calcium storage (from the ER); and inhibition of phosphodiesterases [4]. Nowadays, the most well-recognized mechanism is that caffeine acts in the CNS as a competitor of adenosine in its receptors [4], inhibiting the negative effects that adenosine induces on neurotransmission, excitation, and pain perception [5,6].

Through blocking adenosine receptors (mainly A1 and A2A subtypes) [7], caffeine seems to competitively antagonize their effect and cause an increased release of dopamine, noradrenalin, and glutamate [3]. The adenosine receptor blocking ability of caffeine is visible at low doses as well, like in a single cup of coffee [3]. Resulting in lower pain perception, more sustainable and forceful muscle contraction, and maintaining or increasing the firing rates of motor units, it consequently allows for greater strength production [5]. Caffeine can affect the mobilization of energy substrates during exercise. It has been suggested that caffeine increases the mobilization of free fatty acids by adrenaline (epinephrine) induction, saving glycogen [5]. It is especially beneficial in aerobic training, because lower glycogen utilization results in longer training times [8]. Although many aerobic and anaerobic sports depend on muscle glycogen, athletic performance is dependent on other mechanisms, such as increased calcium mobilization and phosphodiesterase inhibition [5,9].

Through mobilization from intracellular storages, caffeine can induce calcium release from the sarcoplasmic reticulum, and can also inhibit its reuptake. Through these mechanisms, caffeine can increase contractility during submaximal contractions [3]. Hence, caffeine is an ergogenic aid in various exercises, such as endurance sports, exercise efforts with high glycolytic demands, resistance trainings, and racket, combat, and team sports [7].

Caffeine acts as a nonselective competitive inhibitor of phosphodiesterases, enzymes hydrolyzing phosphodiester linkages in molecules, like cyclic adenosine monophosphate (cAMP), inhibiting their degradation. cAMP stimulates lipolysis, but these mechanisms of action require very high doses of caffeine [3].

The role of caffeine in endothelial function was also demonstrated, where caffeine works as a nitric oxide (NO) stimulator, NO inhibitor, and inhibitor of NO second messenger cyclic guanosine monophosphate (cGMP) [10]. It was suggested that the mechanisms of endothelial dysfunction and vascular smooth muscle dysfunction involve inactivation of NO by reactive oxygen species (ROS), inflammation, increases in vasoconstrictors, an increase in the endogenous endothelial NO synthase (eNOS), inhibitor asymmetrical dimethylarginine, and an abnormality of shear stress.

Caffeine has been shown to reduce cerebral blood velocity and celebral blood flow (CBF), assessed using a number of different techniques [11,12]. Despite the observed reduction in CBF, and therefore decreased supply of metabolic substrates, caffeine has consistently been shown to improve reaction times [13,14,15] and alertness [16]. Earlier concerns were that caffeine has an acute diuretic effect and may lead to dehydration, but concerns regarding unwanted fluid loss associated with caffeine consumption are unwarranted, particularly when ingestion precedes exercise [17,18,19].

## 2. Materials and Methods

The aim of the study was to collect available data on scientific publications describing the complexity of effects and mechanisms of caffeine and EDs, and to also describe the consumption of caffeine-containing beverages among children, adolescents and young adults, with factors such as:The positive and adverse effects of different caffeine dosages in the body and in human performance;Short- and long-term effects of EDs in different age groups;Motivations for ED consumption;Effects of EDwA consumption among youngsters;The categories and topics selected for this publication were based on our interest; these topics are very popular in different research areas as well.

The method for collecting data for the work was described earlier by Torres et al. (2020) [20] and it is outlined in Figure 1.

Eligibility Criteria.When reviewing the literature, the following terms were important:
Structure of caffeine, absorption, consumption;Caffeine-containing beverages;Positive and adverse physiological effects of caffeine;In special cases, animal studies were also included in the work.Sources of information.Search for data was accomplished electronically, using mostly PubMed databases. Other scientific sources were also added in case of fulfilling search criteria.Search for information.

Data search and selection was conducted with the aid of keywords, described in Figure 2.

Using the above keywords, altogether 6312 publications were found to fit the search criteria (Figure 2). The number of eligible publications from sources other than PubMed was 185. The earliest papers considered were published in 1982, while the upper time limit was 2021. Only eight publications were selected before the year 2000, and research data published after 2005 were preferred. The number of studies published between the years 2005–2021 was 134, and only data appearing in scientific journals were considered.

1.Selection of Studies

When considering publications and data, the criteria were: the age of the studied population, description of consumed dosages, and measured effects. Publications generalizing former data (e.g., dosage), or research with low sample sizes were excluded.

2.Data Collection

All considered data and publications were grouped in different topics according to their effects, or age groups.

3.Synthesis of Results Data were analyzed with regard to the following topics:

Timing of caffeine ingestion; effects of caffeine consumption in different dosages; products containing caffeine; caffeine, energy drinks (EDs), and physical performance; caffeine and cognition; children, young adults, and EDs; ED consumption with alcohol (EDwA); sense of coherence and depression; and motivations for ED consumption.

## 3. Results

### 3.1. Timing of Caffeine Ingestion

After ingestion, caffeine is quickly absorbed [21] from the gastrointestinal tract into the circulatory system. The peak blood caffeine levels are reached 1 h post-ingestion [7], while others suggest that maximal plasma concentration is reached after 30–60 min from consumption [22]. Maximal plasma concentrations measured between 15 and 120 min after intake have also been reported. These measured time gaps are due to inter-individual differences, and delayed gastric emptying and metabolism [7,23,24]. After absorption, caffeine promptly gets into all the body tissues and also crosses the blood–brain [25], blood–placenta, and blood–testis barriers [3]. The half-life of caffeine in humans ranges from a minimum of 2 to a maximum of 12 h (~3–5 h on average) [21], and around 3–7 h in adults [22].

Caffeine and ED intake in different dosages result in absorption in the gastrointestinal tract. After absorption, the peak concentration of caffeine in the blood depends on many factors, and there are large individual differences. The caffeine content of EDs is different from the labels, since they contain taurine and glucoronolacton, increasing the effects of caffeine; guarana also contains caffeine (Figure 3). According to the label, it consists of 1000 mg taurine, 600 mg glucoronolacton, 80 mg caffeine, 18 mg niacin, 6 mg pantothenic acid (B5 vitamin), 2 mg B6 vitamin, B2 vitamin, B12 vitamin, inozitol, carbonated water, sacharose, glucose, (27 g sugar), citric acid, and caramel flavour [26].

### 3.2. Effects of Caffeine Consumption in Different Dosages

Caffeine can be detected in the blood 5–15 min after consumption, and the peak of its effect was measured at between 40–80 min [21,27]. Caffeine in higher doses (9–13 mg/kg) will not have any additional effect on physical performance, but might increase the side effects of caffeine (Table 1) [5]. Ingestion of high caffeine doses (~10–13 mg/kg) produced side effects of the gastrointestinal system, nervousness, mental confusion, inability to focus, and sleeping problems in some subjects [21,28], while in lower doses (7–10 mg/kg) it produced chills, flushing, nausea, headache, palpitations, and tremors [29,30,31]. Reducing caffeine dosages to a moderate level (5–6 mg/kg), ergogenic effects were maintained, the physiological responses and side effects were also reduced but did not disappear [21,32]. Caffeine dosages of 200 mg or more will induce toxicosis, resulting in signs of nervousness, insomnia, digestion problems, muscle cramps, and periods of unreasonable alertness [33,34]. According to the literature, low or moderate dosages of caffeine consumption (3–6 mg/kg) 60 min before training are recommended to have positive effects [5,9,35]. If a low caffeine dose (3 mg/kg) was administrated, the ergogenic effect of caffeine was measurable without changes in exercise heart rate (HR) and the levels of catecholamines, lactate concentration, free fatty acids (FFA), and glycerol [20,27].

Higher doses of caffeine or 1000 mg/day caused toxic symptoms [31], restlessness, hyperactivity, headaches, nausea, dizziness, trembling, spasm, extrasystole, and tachycardia, while after caffeine consumption of around 2000 mg/day, hospitalization, toxic symptoms and cardiovascular, gastrointestinal, psychological/neurological, and metabolic symptoms might appear [31]. The safe dosage intervals of caffeine intake are still missing, but data in the literature suggest that a healthy adult individual might consume 400 mg of caffeine daily (Table 1) [37]. Data regarding safe consumption and safe dosages for children are lacking [19], although some stated that the recommended limit for children and adolescents should not exceed 100 mg/day [40] or 2.5 mg/kg per day, respectively [19]. Some data suggest that less than 600 mg/day in caffeine intake will have light, temporary, and reversible cardiovascular effects [38,39]. Caffeine is the main active substance of energy drinks (ED) and sports foods [1,41], and excessive consumption might cause anxiety and irritability [42,43] can be toxic, inducing such side effects as tachycardia, vomiting, HR problems, shock, or death [44].

The estimated caffeine exposure from ED or energy shots was calculated for New Zealand children (5–12 years old), teenagers (13–19 years old), and young men (19–24 years old) [45]. After consuming a single retail unit, 70% of children and 40% of the teenagers who consumed caffeine were estimated to have exceeded the adverse-effect level of 3 mg/kg body weight per day beyond their regular dietary exposure. An average child, teenager, or young man would all, on average, exceed the adverse-effect level after consuming a single retail unit of ED/energy shot above their baseline dietary caffeine exposure [45].

Ellison et al. [46] reported that children (6–10 years) ingested caffeine on an average of 8 out of 10 days. Variable caffeine intakes were also reported of up to 16 mg/day by 7- to 8-year-olds, 24 mg/day by 9- to 10-year-olds, and 37.4 mg/day by 5- to 18-year-olds [47]. Symptoms of caffeine withdrawal are very similar to the signs of intoxication.

On the contrary, evidence indicates that caffeine could be recommended for youths with special conditions (ADHD and apnea of prematurity), but not suggested for healthy children, especially in moderate and high doses that reportedly cause physiological alterations [20]. Coffee was associated with a probable decreased risk of breast, colorectal, colon, endometrial, and prostate cancers; cardiovascular disease and mortality; Parkinson’s disease; and type-2 diabetes [48,49]. A number of epidemiological studies confirmed a link between higher coffee consumption levels and better performance on cognitive tests in older adults, and an inverse relationship between coffee consumption and the risk of developing Parkinson’s or Alzheimer’s disease, as well as a lower risk of stroke [49].

There is evidence that caffeine may reduce sensations of pain through its effects on adenosine receptors [49]. In addition, when consumed for its central nervous system stimulant effects, caffeine also possesses pain-relieving properties. Studies conducted in animals and humans have found that the acute administration of caffeine is associated with decreased pain [50]. Caffeine has been linked with migraines for many years, on the one hand as a trigger, and on the other as a cure [49]. Caffeinated headache medications, either alone or in combination with other treatments, are widely used by patients with headache [51]. Compared with analgesic medication alone, combinations of caffeine with analgesic drugs, including acetaminophen, acetylsalicylic acid, and ibuprofen, showed significantly improved efficacy in the treatment of patients with TTH or migraines [51].

### 3.3. Products Containing Caffeine

#### 3.3.1. Energy Drinks

The energy drink (ED) named Red Bull appeared in Australia in 1984, being very popular since then in Europe and in North America as well [52]. EDs are soft drinks, (carbonated) sodas containing caffeine [34], taurin, glucuronolactone [40,48], carbohydrates, different types of vitamins (absorbed differently) [22,50,51,52,53,54,55,56], niacin, pyridoxin, riboflavin (B2), ginseng substrate, inozit (B8), guarana (caffeine, theobromine, and teophylline), Ginkgo biloba extracts [57], herbs [19], and l-carnitin [33,34,58,59]. The actual caffeine content of these beverages depends on the quality and type of processing of the raw materials [3].

Many other caffeine-containing beverages and products exist, including tea, chocolate [60], cocoa, cola nuts [51,61], mate, guarana [19,31,42,62], medications, dietary supplements, and soft drinks [53].

The positive effects of EDs on cognition and physical performance depend on the combination of the different components [63,64]. EDs often contain several plant compounds acting synergistically to increase energy metabolism and lift one’s mood [65]. Caffeine combined with other plant molecules have shown to increase alertness, concentration, mood, and decrease fatigue [33,63,65,66].

#### 3.3.2. Caffeine, EDs, and Physical Performance

Caffeine-containing products are widely consumed among different age groups and for a number of reasons, from socialization to mental and physical alertness [1,23]. After consumption, the responses in people vary, with a range of positive, neutral, or negative effects on performance depending on the person’s genotype, training status, habitual use of caffeine, gender, caffeine source, and age [1].

Caffeine was on the doping list until 2004, and the International Olympic Committee listed it among their prohibited substances since it has cognitive and muscle contraction and motor control effects, thus affecting sports performance. Professional athletes testing positive for more than 12 µg/L of urine—corresponding to about 5–6 cups of coffee in a day—were banned from events like the Olympics [21].

The effectiveness of caffeine and EDs on sports performance mainly depends on three factors: (1) dose and timing of administration; (2) type of sport; and (3) bodily response to caffeine [1,67]. Consumption of caffeine seems to be effective in long-lasting sport activities, with the greatest effects in sports involving fatigue during or toward the end of the event. Caffeine does not seem to be effective in exercises having very high intensity, or a power output lasting seconds, like sprints or lifts [1]. Around 3 to 6 mg/kg of caffeine seems to be the optimal dose for most people, although recently, increased attention appeared in lower (≤3 mg/kg) caffeine doses, suggesting that these doses are also ergogenic [68]. Comparing the influence of caffeine on physiological responses to exercise between boys and men, Turley and colleagues [69] found that boys responded with greater increases in blood pressure after caffeine consumption (5 mg/kg), HR decreased in boys but not in men, and VO2max did not change in either group. In a similar setting with the same dosages, no difference was found between the physiological parameters of boys and girls [69].

EDs containing caffeine have become the most popular beverages in sport settings [70] in both recreational and trained athletes (Table 2) because of their proposed ergogenic effects [8], and are frequently consumed by athletes prior to competitions to improve performance [71]. Testing the effects of Red Bull EDs on psychomotor performance (e.g., reaction times) and physical performance with volunteers, in contrast with the control drink, Red Bull significantly increased aerobic and anaerobic performance [53]. A lower dose of caffeine consumption (12.5–100 mg/day) increased aerobic capacity and improved reaction times [19]. Studies using high doses of caffeine (~10–13 mg/kg) reported ergogenic effects in endurance-type activities, and there were also pronounced effects on the physiological responses to exercise, like increased HR, a doubling of catecholamine levels, higher BL levels, and increased free FFA and glycerol levels in the blood of many subjects [21]. In healthy, physically active females consuming high dosages of EDs, no change was measured in mood, alertness, and concentration [65]. In boys, 3 mg/kg and 5 mg/kg of caffeine increased handgrip strength, where the lower dose increased peak power, while the higher improved the mean power of the subjects [72].

Only a few studies have shown gender differences in the effect of caffeine supplementation on sports performance, and their results are controversial [5]. Chewing caffeine gum (3–4 mg/kg) improved mean and sprint performance power in the final 10 km of a 30 km trail in male and female cyclists, most likely through an increase in CNS activation [5,73]. Acute caffeine ingestion (6 mg/kg) increased HR and BL levels during exercise in the heat, but it had no impact on thermoregulation or endurance capacity in either gender. Under exercise-heat stress, caffeine reduced ratings of perceived exertion (RPE) and fatigue in males, but not in females [5,74]. Ingestion of 3 mg/kg caffeine enhanced endurance performance in women [75]. The magnitude of performance enhancement observed was similar to that of men, despite the significantly greater plasma caffeine concentrations after exercise. These results suggest that current recommendations for caffeine intake (i.e., 3–6 mg/kg) before exercise enhances endurance performance, although data were derived almost exclusively from studies of men, but it may also be applicable to women [75]. In this respect, studies in the general population have already shown that the stimulating effects (less drowsiness and higher activation) of caffeine are greater in men than in women [5,76].

Pedersen and coworkers [77] described that caffeine (8 mg/kg body weight), coingested with carbohydrates by well-trained athletes, was responsible for higher rates of post-exercise muscle glycogen storing in comparison to the ingestion of carbohydrates alone after depletion of glycogen stores [77].

In patients with coronary artery disease, in exercise-stress tests, caffeine at a dose of 250 mg had no effect on exercise duration, time to onset of angina, and time to onset of ST-segment depression, although peak blood pressure increased by 7 mmHg [78]. Another study suggested that regular intake of caffeinated beverages could provide protection against the risk of cardiovascular disease mortality in nonhypertensive elderly patients [79].

In the elite sports of university students, caffeine supplementation (6 mg/kg), compared to the placebo, significantly increased maximal voluntary isometric contractions (MVIC; 5.9%) and submaximal voluntary isometric contractions (T (lim; 15.5%). The ergogenic effect of caffeine on muscle power and muscle endurance did not show gender differences [5,80].

The effect of EDs on the trunk muscles was measured in healthy males with sit-up tests until exhaustion. Performance after consuming EDs increased by 13.2%, while in the placebo group it decreased by 0.7% half an hour after consumption [81]. Kammerer and coworkers [56] studied the fitness and cognitive performance of male solders in a double-blinded, randomized, placebo-controlled study. All the subjects took part in a spiroergometry test, measuring maximal oxygen consumption (VO2max), maximal HR (HRmax), and the time to exhaustion (TTE). Muscle performance was measured with hand-grip strength tests in both hands, and lower limb strength was measured with high jumps (time spent in the air). Cognitive attention was measured with the aid of attention (Grid) and WAIS tests. No significant differences were found in VO2max, HRmax and TTE, hand grip strength, high jump, concentration, short-term memory after EDs, or placebo consumption. Young male sportsmen who consumed EDs (e.g., Red Bull, Hype) or a placebo and their effects in the cardiovascular and pulmonal system were tested [8]. Changes in metabolism were detected with BL levels. Subjects were tested three times after consuming the drinks randomly (40 min before testing) with a spiroergometric protocol. Both EDs increased the VO2max levels and the time to exhaustion (Red Bull, 11.5%; Hype, 9.9%) compared to the placebo, but did not influence the HR and BL level. The effects of the Red Bull ED on repeated sprint performances was also studied in female soccer players [71]. One hour before the sprint test, subjects drank 225 mL of Red Bull (1.3 mg/kg caffeine). The mean sprint time was similar before and after ED consumption and caffeine did not change the HR and RPE either. In another study, volunteers were tested concerning changes of VO2max and the ratings of RPE with the aid of a Borg scale after ED consumption. Subjects drank EDs (three types) or a placebo one hour before training. The training itself lasted 15 min on a treadmill at 70% intensity of their earlier measured VO2max. ED consumption did not increase VO2max or RPE levels compared to the placebo [82].

The effects of caffeinated EDs (1 and 3 mg/kg) were studied in the case of muscle performance, measured by half squats and bench presses. Subjects first consumed 3 mg/kg; the second time, they consumed 1 mg/kg of caffeine-containing EDs; the third time, they were offered a placebo 60 min before the exercise. Maximal muscle performance (1RM) was measured before testing [70]. According to the obtained data, 1 mg/kg caffeine did not improve muscle performance in the range of 10–100% of 1RM, while 3 mg/kg caffeine consumption increased performance for the half squat (7%) and the bench press (7%) also compared to the placebo. Thus, 1 mg/kg caffeine did not have an ergogenic effect on muscle performance.

University students’ (10 male, 10 female) performances were measured by testing their reaction time in hearing and isometric handgrip strength tests after ED (Red Bull) and control (placebo) consumption. Subjects were all right-handed and always tested in the sitting position 1 h after ED consumption. They were pushing a button right after hearing the sound stimulus (10 stimuli were used) with closed eyes. Handgrip strength was tested with the dominant hand. Both the control and ED improved reaction times, but none had an effect on muscle function. The ED by itself was not found to be more effective than the control drink [83].

#### 3.3.3. Caffeine and Cognition

Caffeine seems to prevent or restore memory impairment due to disturbances in brain homeostasis [84], although the cognition-enhancing properties are still a matter of debate [85,86]. Interestingly, consumers of moderate to high levels of caffeine develop a tolerance to caffeine, and only low or non-consumers benefit from an acute administration [87]. Caffeine’s ability to boost cognitive function is widely accepted [88], and moderate doses of caffeine and caffeine-containing EDs have been shown to improve attention, reaction times, improve memory, facilitate vigilance, and improve verbal reasoning [63,85,89,90,91]. However, not all aspects of cognitive performance are enhanced by caffeine. Consumption of caffeinated beverages can impair or have no influence on performance of some cognitive tasks in college-aged students [92,93,94]. Galéra and coworkers [95] reported a significant negative correlation between excess caffeine consumption during pregnancy and the Intelligence Quotient (IQ) in exposed children, supporting current guidelines not to exceed 200 mg of caffeine/day. Thus, caffeine likely enhances cognition via its action on general arousal levels.

Animals treated with acute taurine showed reduced fear, while chronically treated mice showed increased fear compared with controls. The authors also reported increased pain sensitivity in the chronically dosed animals, which is consistent with the findings of Serrano et al. [96] in aged CD-1 male mice.

In a study of 9- to 11-year-olds with habitual (109 mg/day) and low (12 mg/day) caffeine consumption, at 50 mg of caffeine after overnight abstention, habitual caffeine users reported a reversal in withdrawal symptoms (e.g., headaches and dulled cognition). Children who did not habitually consume caffeine had no reported changes in cognitive performance, alertness, or headache occurrence [97].

#### 3.3.4. Children, Young Adults, and EDs

The European Food Safety Authority [98] initiated a study to gather data of ED consumption in 16 countries of the European Union. A total of 68% of adolescents (aged 10–18 years old), 30% of adults, and 18% of children (<10 years old) were found to consume EDs. Among adolescents, consumption varied from 48% in Greece to 82% in the Czech Republic, while among children, it varied from 6% in Hungary to 40% in the Czech Republic. The average consumption was 21 in adolescents and 0.49 l in children [57,99]. Overall, 73.6% of respondents reported ever having consumed EDs (12–14 years: 57.0%; 15–17 years: 69.4%; 18–19 years: 77.9%; 20–24 years: 83.4%) [100].

A representative survey in Australia has shown that elementary and high school populations consume significant amounts of caffeinated drinks, including EDs [101]. Adolescents were found to consume it from their 10th year on average, with 56% admitting lifetime consumption between 12–18 years [102]. In the UK, the rate of consumption of EDs grew by 155% between 2006 and 2014. Young people were found to consume more EDs (3.1 per month) than their continental counterparts (2.1/month) [103]. In the U.S., EDs are the second most common dietary supplement used by young people [34]. A large sample demonstrated that 30% of high school students admitted to being regular consumers [104], and there is a strong correlation between ED consumption and smoking, alcohol consumption, and drug abuse [105]. Numerous studies have pointed out the adverse health effects of EDs and their connection to destructive behaviors [103]. ED consumption is also widespread among American teenagers (13–17 years), and the rate depends on demographic, psychosocial, lifestyle, and substance abuse factors [106]. In a representative American study, 40% of adolescents reported daily consumption [104]. There is a strong correlation between the consumption of EDs and increased soft drug use, which in turn is linked to increasing drug abuse in general [107]. Consumption of EDs among adolescents is also associated with other potentially negative health and behavioral outcomes, such as sensation-seeking, the use of tobacco and other harmful substances, and binge-drinking, as well as a greater risk for depression and injuries that require medical treatment [57,108,109]. Surveys of American high schoolers indicated a strong link between ED consumption and hyperactivity or a general lack of attentiveness [110]. A study of 15- to 16-year-olds demonstrated a strong correlation between caffeine consumption, aggressive behavior, mood disorders [19,42] and other behavioral disorders [57,111]. Problems with cognitive capabilities were also reported [57,112].

Many other studies report on the adverse effects of ED consumption [34,113]. In a Finnish sample (12–18 years), there was a strong correlation between daily ED consumption and symptoms such as headaches, sleep disturbances, and fatigue [113]. The known side effects of excessive caffeine consumption include tachycardia, tremors, high blood pressure, and in the most serious cases, sudden death [30,32,114]. Among proven negative consequences of caffeine consumption in children and adolescents, effects on the neurological and cardiovascular systems have been found, which in turn can cause physical dependence and addiction [57,115]. Recent studies described significant hemodynamic changes in healthy young individuals following ED consumption, with elevated systolic and diastolic BP, increased cardiac output and myocardial load, repolarization abnormalities and reduced cerebral blood flow velocity [116,117]. A significant increase was also found in circulating catecholamines, reflecting sympathetic activation [118], since caffeine stimulates the CNS and cardiac systems [44,119]. Among Icelandic children (10–12 years), stomach pains and headaches as well as insomnia were more common among ED consumers [120]. ED use may also be accompanied by anxiety, nervousness, migraines, gastrointestinal disease, metabolic acidosis, insomnia, arrhythmia, chest pain, and other cardiovascular complications [121]. Park and coworkers [122], who studied 12- to 18-year-olds, stated that ED consumption was significantly correlated with poor sleep, increased stress, depression, and suicidal thoughts.

Reissig and colleagues [42] reported on levels of caffeine ranging from 50 to 505 mg, and suggested that the recommended limit for children is 100 mg per day [123]. Thomson and Scheiss [124] described that a single ED could push 70% of children and 40% of teenagers past the level of adverse effects (3 mg/kg/day) when combined with other dietary sources. Accidental consumption was also found to be frequent. About half of ED-related calls to the U.S. National Poison Data System between the fall of 2010 and 2011 involved children under the age of 6 years [125].

There are many common ingredients, other than caffeine, found in the most popular EDs. McLellan and Lieberman [126] described amounts of the non-essential amino acid taurine to be between 750 and 1000 mg/serving. The normal diet typically contains 40 to 400 mg/day [127], and no more than 3000 mg/day is recommended [128]. Taurine can modulate calcium release, so there are potential impacts on the brain, heart, and skeletal muscle [125,129]. Cardiac effects are exacerbated when taurine and caffeine are ingested together [130], since caffeine alone can increase BP and HR. EDs have no therapeutic benefit, and both the known and unknown pharmacology of various ingredients, combined with reports of toxicity, suggest that these drinks may put some children at risk of serious adverse health effects [21].

Additionally, 75.4% of the ED consumers were young (10 to 17 years; Figure 4) [131].

Data show that boys consume more EDs (higher possibilities and higher dosage, 60.3%) than girls (39.7%) [103,109,120,131,132,133,134,135], but in girls, the same ED consumption causes more unpleasant symptoms. Women did not tolerate exaggerated caffeine intake as well as men [131]. Among Norwegian middle school students, boys were drinking double the amount of EDs than girls [118]. Among adolescents, proportionately more boys (41.5%) than girls (26.3%) drank EDs daily [136]. In a Belgian sample, parents of 11- to 20-year-olds reported that their children consumed EDs more than once weekly (boys 14%, girls 7.6%) [137].

A strong correlation was found between the quantity of consumption and the increase of unpleasant symptoms (*p* < 0.001) [130]. A total of 67.1% experienced unpleasant symptoms, where 42.7% complained about more than one side effect, like rapid HR (51%), insomnia (38.8%), weakness, shivering (25.4%), headaches (21.4%), and in some cases, even loss of consciousness [131].

According to Visram and colleagues, those who consumed EDs several times a day were 4.5 times as likely to experience headaches and 3.5 times as likely to experience sleeping problems, in comparison with those not consuming these drinks [103].

In another study, 71.4% of respondents reported experiencing adverse effects of EDs, and 10.2% simultaneously experienced four or more of the symptoms on the list. There was no significant difference between male (19.5 ± 2.0 years old) and female (19.2 ± 2.0 years old) respondents in the symptoms experienced. In both sexes, the primary side effects were tachycardia (M 32%, F 38%), insomnia (M 26%, F 38%), and tremors (M28%, F30%) [132].

Symptoms were significantly related to whether the EDs were consumed by themselves or mixed with alcohol [130]. These symptoms or syndromes occurred among 86.5% of the subjects, who were mixing EDwA (Table 3) [128]. Gradvohl and colleagues [138] conducted a survey among university students in 2013 showing that consumers mixing alcohol with ED were more likely to drink more alcohol both at parties and on ordinary days, and that they took part in binge-drinking more frequently than those who only consumed alcohol.

### 3.4. ED Consumption with Alcohol (EDwA)

High consumption of EDs and caffeine among teenagers and young adults raise serious concerns about their adverse effects in the CNS, especially when combined with alcohol consumption. Elevated rates of binge-drinking and risks of alcohol dependence have been associated with alcohol mixed with energy drinks (EDwA) versus alcohol alone. Because co-ingestion of EDs and alcohol is frequent, and alcohol may disrupt the timing of pubertal development in females, ingestion of EDwA in early adolescence may cause pubertal delays [22,140,141]. It has been suggested that subjective mood change and cardiovascular responses to caffeine are dependent on the gender and pubertal stage [22,142,143].

Results from laboratory studies indicate that when EDs (or caffeine) are ingested with alcohol, the desire to drink more alcohol is more pronounced in both humans and animals than with the same alcohol dose alone [144,145,146]. It seems to be a “grave danger” that adolescents are drinking more alcohol than intended, and that they are more likely to drive after drinking alcohol mixed with EDs [147]. It has been supposed that caffeine can counteract the sedative effects of alcohol and, therefore, drinkers may not feel the symptoms of alcohol intoxication [22]. Mixing EDwA is popular among teenagers and college students [148], bringing about further high-risk behaviors, such as excessive alcohol consumption, smoking, and drug abuse [149]. A significant correlation has been found between EDwA and smoking, alcohol, and cannabis consumption [139]. Those who drank EDwA were also more likely to use marijuana, ecstasy, and cocaine [150]. In an Italian sample of 30,588 high school students, (15–19 years), 41.4% and 23.2% of respondents reported drinking EDs and EDwA [22,139], while in a Hungarian population, 24% of consumers drank EDs alongside or mixed with alcohol [132]. In another study, 23.2% of high school students (15–19 years) admitted consuming EDwA [139], and young regular consumers of EDs were also more likely to consume larger quantities of alcohol per occasion, and were also more prone to make poor decisions (e.g., drunk driving) or engage in aggressive behavior [106]. The same young people were also at increased risk for excessive alcohol consumption later in life [132,151].

In animal models [152] testing binge-drinking, treating young male rats with alcohol, EDs, or EDwA for 6 days, alcohol alone or EDwA, impaired performance in a novel object recognition task was found. Rats given ED alone had no statistically significant lower discrimination index compared to the controls. There were similar findings in a social recognition test. What raised concerns for human health was the finding that rats treated with a combination of EDwA showed a significantly greater preference for alcohol in a conditioned-place-preference test. A potential mechanism could be the additive effects of taurine and alcohol on dopamine release in the nucleus accumbens and stimulation of the brain’s reward circuitry, as demonstrated by Ericson et al. [153].

A similar study was performed by Krahe et al. [154] using P40 male and female Swiss mice. Mice receiving the combination of EDwA had significantly higher locomotor activity, increased anxiety, and shorter latencies to fall off the rotarod. These findings also suggest that taurine and caffeine in EDs are not protective against the effects of alcohol, and instead exacerbate them.

### 3.5. Sence of Coherence and Depression

According to the results of Tóth et al. [132], both a weak sense of coherence and a tendency to become depressed increased the chances of addiction, while a strong sense of coherence diminished the effects of depression. If motivations of consumption were analyzed, significant differences were found between those consuming EDs daily and less frequently. Even considering sex and age, more frequent consumers marked taste, stimulation, thirst, and work-out much more often than subjects of the infrequent consumer group. Therefore, those who consumed EDs mostly for the reasons listed above were more likely to become addicted [132]. Young people with higher academic averages, a higher sense of coherence, higher levels of parental monitoring, and who had more educated parents were less likely to consume EDs [103].

### 3.6. Motivations for ED Consumption

In the case of motivations for ED consumption, the most frequent responses were because of taste (50.2%), to feel energized (12.7%), to mix with alcohol (19.3%), or to stay awake (11%), and 5.3% reported drinking out of curiosity [136]. According to several studies [103], the primary motivations for ED consumption were taste and energy/stimulation. Many users consumed EDs because of sleep deprivation (67%), to be energized (65%), and alongside alcoholic drinks at parties (54%) [155]. EDs have been increasingly consumed, especially among adolescents, to sustain alertness and boost energy [43,109], and they also often consumed EDs just to enjoy their taste or celebrate special occasions with alcohol [43,156]. If they were asked how often they mixed EDwA and the reasons for doing so, answers were for better taste, less alcohol side-effects, social reasons, curiosity, to consume more alcohol, or to feel alert [136].

In a Hungarian sample asking respondents about their motivations for ED consumption, their primary choice was fatigue, followed by taste [132]. Motivations in the case of male and female participants did not differ significantly, but males were more likely to use EDs for fun or before work-outs, while females mostly consumed them to fight fatigue. Subjects were much more likely to consume EDs if their parents, siblings, or friends were also consumers [132].

Those participating in sports, especially males, used EDs to improve sports performance [105], in many cases prior to competitions, and students especially while studying for exams. EDs are also commonly consumed at dance parties, as they require sustained energy for prolonged activity into the late hours of the night [21].

## 4. Conclusions

Although caffeine and EDs are often used to increase performance, data in the literature are controversial in terms of the positive physical and cognitive effects of these products. These effects strongly depend on dosages, ages, sexes, the time of consumption, and personal sensitivity. EDs are widely used among youth, adolescents, and even among children, although many of them admit unpleasant side effects, like weakness, shivering, headache, tachycardia, insomnia, tremors, or depression. The ratio of unpleasant symptoms is much higher among those drinking EDwA. Consumption of EDs is correlated with depression, negative behavioral changes, and they are addictive. Consumers of EDwA often use drugs or become binge-drinkers. Further detailed studies are needed to describe dosages having positive effects in adults, and stricter compound descriptions, as well as dosage and regularity details of ED usage are needed to protect developing children and adolescents from the burdens of addiction and of negative neurological or behavioral changes. Understanding the unforeseen consequences of ED and EDwA consumption in young people would be very important to create stricter regulations in advertisements and in the sale of these beverages. The long-lasting effects of addiction are the responsibility of parents, trainers, and societies.

The population most in danger among all consumers is definitely children, adolescents, and young adults, since they consume the largest quantity of EDs, consequently having the highest occurrence of adverse effects and symptoms. The solution would be stricter regulations for distribution and retailer possibilities. For children and adolescents, neither caffeine nor ED consumption is recommended. For young adults, low (≥3 mg/kg) or moderate (3–6 mg/kg) dosages are accepted without adverse side effects.

Caffeine and ED consumption is not recommended at all for children and adolescents at any dosage. For young adults, low (≥3 mg/kg) or moderate (3–6 mg/kg) caffeine dosages are acceptable or recommended.

## Figures and Tables

**Figure 1 ijerph-18-12389-f001:**
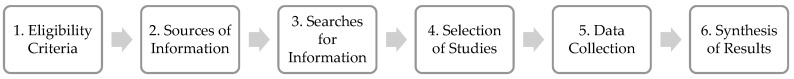
Steps for the review of the literature (Torres et al., 2020) [20].

**Figure 2 ijerph-18-12389-f002:**
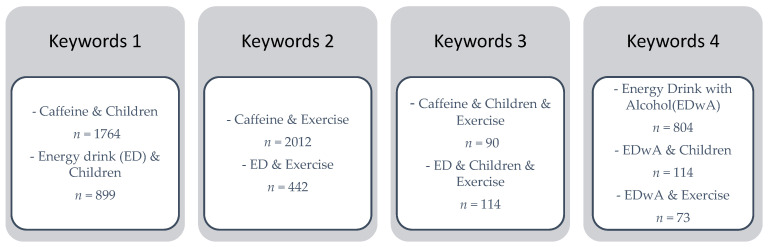
Keywords used when searching data.

**Figure 3 ijerph-18-12389-f003:**
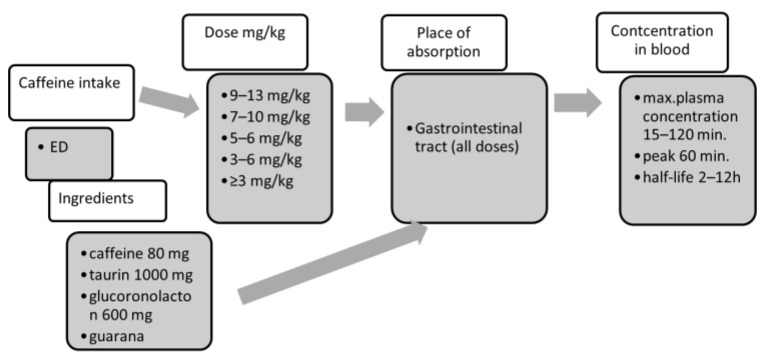
Caffeine absorption.

**Figure 4 ijerph-18-12389-f004:**
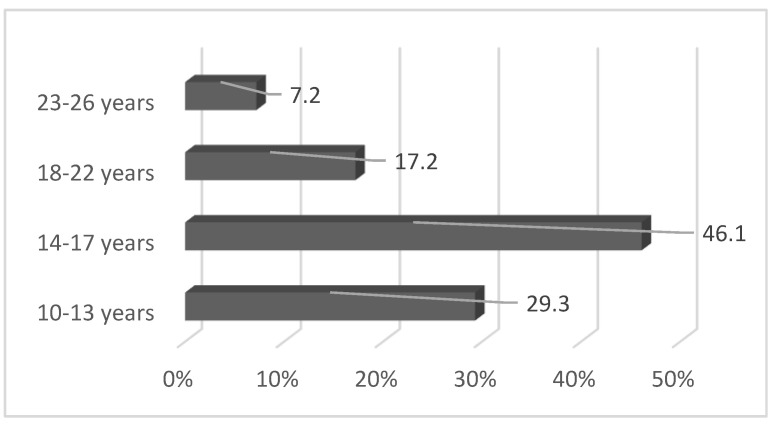
Age distribution of ED consumers in a Hungarian sample of 1459 subjects [131].

**Table 1 ijerph-18-12389-t001:** Effects of caffeine consumption in different dosages.

Authors	Dose (mg/day)	Effects
Kaplan et al., 1997 [29] Willson, 2018 [31] Smith, 2002 [36]	250 mg	increased arousal, alertness, concentration, well-being
Kaplan et al., 1997 [29] Willson, 2018 [31]	500 mg	increase nervousness, anxiety, excitement, irritability, nausea, paresthesia, tremor, perspiration, palpitations, restlessness, possibly dizziness
Higgins-Babu, 2013 [37]	400 mg/day	safe dose for adults
Nowak-Goslinski, 2019 [38] Turnbull et al., 2017 [39]	~600 mg/day	reversible cardiovascular effects
Bedi et al., 2014 [33] Alsunni, 2015 [34]	200 mg	nervousness, insomnia, problems of digestion, muscle cramps, and periods of unreasonable alertness muscle cramps, and periods of unreasonable alertness
Willson, 2018 [31]	≤1000 mg/day	toxic symptoms hyperactivity, headaches, nausea, dizyness, trembling, spasm, extrasystole, tachycardia
Willson, 2018 [31]	~2000 mg/day	toxic symptoms, requires hospitalization, ventricular fibrillation cardiovascular symptoms
Willson, 2018 [31]	~3000 mg/day	lethal
Authors	Dose (mg/kg)	Effects
Mielgo-Ayuso et al., 2019. [5] Pickering-Kiely 2018 [35] Goldstein et al., 2010 [9]	3–6 mg/kg	positive effects increase physical performance
Mielgo-Ayuso et al., 2019 [5]	9–13 mg/kg	no positive effect in physical performance
Graham et al., 1995 [28] Spiret, 2014 [21]	~10–13 mg/kg	troubling side effects of gastrointestinal upset, nervousness, mental confusion, inability to focus, and disturbed sleeping
Kaplan et al., 1997 [29] Kerrigan-Lingsey, 2005 [30] Willson, 2018 [31]	~7–10 mg/kg	chills, flushing, nausea, headache, palpitations and tremor
Graham et al., 1995 [28] Spiret, 2014 [21]	3 mg/kg	no negative effect in physiological responses

**Table 2 ijerph-18-12389-t002:** Effects of EDs in physical activity.

Authors	N	Dose (mg/day)	Effects
Rashti et al., 2009 [65]	*n* = 10	ED (230 mg)	no change in mood alertness and concentration
Alford et al., 2001 [53]		not given	increased aerobic and anaerobic performance
Seifert et al., 2011 [19]	not given	12.5–100 mg/day	increased aerobic capacity improved reaction time
Authors		Dose (mg/kg)	Effects
Mielgo-Ayuso et al., 2019 [5]; Pickering- Kiely, 2018 [35]; Goldstein et al., 2010 [9]	*n* = 20	3–6 mg/kg	positive effects increased physical performance
Mielgo-Ayuso et al., 2019 [5]	not given	9–13 mg/kg	no positive effect in physical performance
Spiret, 2014 [21]	not given	~10–13 mg/kg	ergogenic effects in endurance-type activities increased heart rates higher blood lactate levels
Paton et al., 2015 [73] Mielgo-Ayuso, 2019 [5]	*n* = 20	3–4 mg/kg	improves mean and sprint performance power in male and female cyclists
Suvi et al., 2016 [74] Mielgo-Ayuso, 2019 [5]	*n* = 23	6 mg/kg	increases HR and blood lactate reduces ratings of perceived exertion and fatigue in males no positive effect in endurance capacity
Skinner et al., 2019 [75]	*n* = 27	3 mg/kg	enhanced endurance exercise performance in women
Chen et al., 2015 [80] Mielgo-Ayuso et al., 2019 [5]	*n* = 20	6 mg/kg	ergogenic effect of caffeine on muscle power and muscle endurance

**Table 3 ijerph-18-12389-t003:** Some data showing the effects of energy drinks with alcohol consumption among adolescents or young people.

Authors	Country/*n*	Years	ED/EDwA	Significant Correlations with Different Symptoms/Syndromes
Huhtinen et al., 2013 [113]	Iceland *n* = 11,267	10–12 years	ED	headaches, sleep disturbances, fatigue
Kristjansson et al., 2014 [120]	Finnland *n* = 5840 in 2007	12–18 years	ED	stomach pains headaches insomnia
Gradvohl et al., 2015 [138]	Hungary *n* = 1066 in 2013	students 18–24 years	EDwA	Binge drinking
Park et al., 2016 [122]	Korea *n* = 68,043	12–18 years	ED	sleep dissatisfaction severe stress depressive mood suicide attempts
Soós et al., 2016 [131]	Hungary *n* = 1495	10–26 years	ED EDwA	rapid HR insomnia weakness, shiver headache
Kim et al., 2017 [43]	Korea *n* = 121,106 in 2014–2015	13–18 years	ED	stress inadequate sleep low school performance suicide attempts
Scalase et al., 2017 [139]	Italy *n* = 30,588 in 2016	15–18 years	ED EDwA	daily smoking binge drinking use of cannabis and other psychotropic drugs

## Data Availability

The data presented in this study are available on request from the corresponding author. The data are not publicly available due to subjects request.

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
