# Peer review of "Effects of Caffeine and Caffeinated Beverages in Children, Adolescents and Young Adults: Short Review"

_ijerph, 2021, doi:10.3390/ijerph182312389_

Round 1
Reviewer 1 Report
This mini review about ED and caffeine intake also contains new data. I would include a description of the study in Hungarian population in the abstract. Thus this mini review could be rephrased as a small study removing some citations, reducing the lengh of the text with citations of poor studies. Would be a short text with more clear remarks.
Minor changes:
- Describe early in the text these abbreviations: ED , CBF, NO and IQ
-Line 29: describe region of absortion of cafeine. Include a new Figure showing a ED with XX mg of cafeine, place of absortion, concentration in blood (Depending on the initial ED content) tissue with AR.
-Figure1. , box 3 of the scheme: replace searces for sources
-Table 1. Order doses by mg/day and mg/kg, remove g/day
-Scheme cafeine effect on AR and localization of AR in brain, vascular tissue and heart
-3.3.4 Colour map depending on cafeine consumption to facilita reader comprenhension
Figure 3. Is hungarian a representative population of EU?America?
-Remove Studies with Low evidence, poor quality or citations stating things like “accompained by seizures, hallucinations, etc” should be avoided. The type of study used to obtain conclusions and the number of participants should be described. Format of table 2 adapted to format of table 3(include number of subjects).
-Conclusions: include a take-home mesage with less general wording. Recomended doses, risk population and future landscape and guidelines.
Author Response
This mini review about ED and caffeine intake also contains new data. I would include a description of the study in Hungarian population in the abstract. Thus this mini review could be rephrased as a small study removing some citations, reducing the lengh of the text with citations of poor studies. Would be a short text with more clear remarks.
Thank you very much for your suggestion! Considering reliable data, we chose
Minor changes:
- Describe early in the text these abbreviations: ED , CBF, NO and IQ
We have describe early in the text these abbreviations: energy drink (ED), cerebral blood flow (CBF), nitric oxide (NO), Intelligence Quotient (IQ). We have indicated the deficiency in the abbreviations.
-Line 29: describe region of absortion of cafeine. Include a new Figure showing a ED with XX mg of cafeine, place of absortion, concentration in blood (Depending on the initial ED content) tissue with AR.
We have a new Figure included.
Caffeine and ED intake in different dosages result in absorption in the gastrointestinal tract. After absorption the peak concentration of caffeine in blood depends on many factors, individual differences are big. The caffeine content of EDs is different from the labels, since they contain taurine, glucoronolacton increasing the effects of caffeine and guarana contains caffeine also. According to the label it consists of 1000mg taurine, 600mg glucoronolacton, 80mg caffeine, 18mg niacin, 6mg pantothenic acid (B5 vitamin), 2mg B6 vitamin, B2 vitamin, B12 vitamin, inozitol, carbonated water, sacharose, glucose, (27g sugar), citric acid, taste and caramel (Laquale, 2007).
-Figure1. , box 3 of the scheme: replace searces for sources
Thank you very much for the thorough reading! We corrected searces for searches. This was a typing/spelling error.
-Table 1. Order doses by mg/day and mg/kg, remove g/day
We removed on Table 1. doses g/day
-Scheme cafeine effect on AR and localization of AR in brain, vascular tissue and heart
Thank you for your suggestion! After reconsidering the aims of this mini review, we decided not to include a scheme explaining mechanisms, effects of receptor functions, cytological properties of AR in any tissue. It would need a long chapter in the text, not simplifying the understanding of the min goals of this paper.
-3.3.4 Colour map depending on cafeine consumption to facilita reader comprenhension
Thank you very much for your suggestion! We think available data are found among retailers, but creating a complete map of consumption would need another full study.
Figure 3. Is hungarian a representative population of EU?America?
Hungarian population is representative in that it brings the European average.
-Remove Studies with Low evidence, poor quality or citations stating things like “accompained by seizures, hallucinations, etc” should be avoided.
We removed these studies.
The type of study used to obtain conclusions and the number of participants should be described.
Format of table 2 adapted to format of table 3(include number of subjects).
We included subject numbers in table 2, as it was published in the publications.
-Conclusions: include a take-home mesage with less general wording. Recomended doses, risk population and future landscape and guidelines.
The population in grave danger of ED consumers are especially children, adolescents and young adults, since both the increase of consumption and unpleasant side effects are reported frequently among them. For children neither caffeine nor ED consumption in recommended. For young adults low (≥3mg/kg), or a bit higher (3-6mg/kg) dosages are accepted without adverse side effects.
Understanding the unforeseen consequences of ED, EDwA consumption in youngsters would be very important to create more strict regulations in advertisements, selling of these beverages. Long lasting effects of addiction are the responsibilities of parents, trainers, societies.
Reviewer 2 Report
The paper properly describes many details of unpleasant side effects, their severity and motivations for consuming ED.
The methods are adequately described.
It is necessary to explain how authors select the categories of results from 3.1 Timing of Caffeine ingestion to 3.6. Motivations for ED Consumption such as using content analysis.
In Conclusions, the authors should discuss the unforeseen consequences and long lasting effects of ED Consumption in detail, as well as comment on implications of the results.
Author Response
The paper properly describes many details of unpleasant side effects, their severity and motivations for consuming ED.
The methods are adequately described.
Thank you very much for your detailed work and opinion!
It is necessary to explain how authors select the categories of results from 3.1 Timing of Caffeine ingestion to 3.6. Motivations for ED Consumption such as using content analysis.
Categories selected for this publication are based on our interest, research topics, and because these topics are very popular in different research areas as well.
The aim of the study was to collect available data of scientific publications describing the complexity of effects and mechanisms of caffeine and EDs. Describing consumption of caffeine containing beverages among children, adolescents and young adults
- positive and adverse effects of different caffeine dosages in the body and in human performance
- short time and long time effects of EDs in different age groups
- motivations for ED consumption
- effects of EDwA consumption among youngsters
In Conclusions, the authors should discuss the unforeseen consequences and long lasting effects of ED Consumption in detail, as well as comment on implications of the results.
Thank you very much for your suggestion!
We believe that the cited, analyzed data in this paper describe well the short and long term effect of EDs. Few thoughts are mentioned in the Conclusions concerning these implications.